# Robustness of Radiomic Features: Two-Dimensional versus Three-Dimensional MRI-Based Feature Reproducibility in Lipomatous Soft-Tissue Tumors

**DOI:** 10.3390/diagnostics13020258

**Published:** 2023-01-10

**Authors:** Narumol Sudjai, Palanan Siriwanarangsun, Nittaya Lektrakul, Pairash Saiviroonporn, Sorranart Maungsomboon, Rapin Phimolsarnti, Apichat Asavamongkolkul, Chandhanarat Chandhanayingyong

**Affiliations:** 1Department of Orthopaedic Surgery, Faculty of Medicine Siriraj Hospital, Mahidol University, Bangkok 10700, Thailand; 2Department of Radiology, Faculty of Medicine Siriraj Hospital, Mahidol University, Bangkok 10700, Thailand; 3Department of Pathology, Faculty of Medicine Siriraj Hospital, Mahidol University, Bangkok 10700, Thailand

**Keywords:** feature reproducibility, lipomatous soft-tissue tumors, T1-weighted magnetic-resonance imaging, tumor segmentation, radiomics

## Abstract

This retrospective study aimed to compare the intra- and inter-observer manual-segmentation variability in the feature reproducibility between two-dimensional (2D) and three-dimensional (3D) magnetic-resonance imaging (MRI)-based radiomic features. The study included patients with lipomatous soft-tissue tumors that were diagnosed with histopathology and underwent MRI scans. Tumor segmentation based on the 2D and 3D MRI images was performed by two observers to assess the intra- and inter-observer variability. In both the 2D and the 3D segmentations, the radiomic features were extracted from the normalized images. Regarding the stability of the features, the intraclass correlation coefficient (ICC) was used to evaluate the intra- and inter-observer segmentation variability. Features with ICC > 0.75 were considered reproducible. The degree of feature robustness was classified as low, moderate, or high. Additionally, we compared the efficacy of 2D and 3D contour-focused segmentation in terms of the effects of the stable feature rate, sensitivity, specificity, and diagnostic accuracy of machine learning on the reproducible features. In total, 93 and 107 features were extracted from the 2D and 3D images, respectively. Only 35 features from the 2D images and 63 features from the 3D images were reproducible. The stable feature rate for the 3D segmentation was more significant than for the 2D segmentation (58.9% vs. 37.6%, *p* = 0.002). The majority of the features for the 3D segmentation had moderate-to-high robustness, while 40.9% of the features for the 2D segmentation had low robustness. The diagnostic accuracy of the machine-learning model for the 2D segmentation was close to that for the 3D segmentation (88% vs. 90%). In both the 2D and the 3D segmentation, the specificity values were equal to 100%. However, the sensitivity for the 2D segmentation was lower than for the 3D segmentation (75% vs. 83%). For the 2D + 3D radiomic features, the model achieved a diagnostic accuracy of 87% (sensitivity, 100%, and specificity, 80%). Both 2D and 3D MRI-based radiomic features of lipomatous soft-tissue tumors are reproducible. With a higher stable feature rate, 3D contour-focused segmentation should be selected for the feature-extraction process.

## 1. Introduction

Lipomatous soft-tissue tumors are a group of tumors that exhibit a variety of clinical behaviors. Lipomas are the most common soft-tissue tumors, accounting for one-third of all soft-tissue tumors [1,2]. Lipomas are benign and they can be treated conservatively with observation only. Surgical excision is necessary when a patient is symptomatic [3]. Local recurrence may occur if the surgical margin is not clear, but the likelihood is very low. Intramuscular (IM) lipomas that are larger than 5 cm, deep-seated, and symptomatic can sometimes be difficult to distinguish from atypical lipomatous tumors (ALTs) or well-differentiated liposarcomas (WDLSs). The World Health Organization (WHO) uses the terms ALT and WDLS to represent tumors with identical histologies but different anatomical locations and clinical outcomes [4]. Atypical lipomatous tumors are located in the extremities or the upper trunk and WDLSs are retroperitoneal or mediastinal lesions. Atypical lipomatous tumors and well-differentiated liposarcomas are low-grade, malignant, adipocytic tumors that recur locally or dedifferentiate to high-grade sarcomas, but rarely metastasize [5,6,7,8]. The gold standard for the diagnosis of ALTs/WDLSs is histopathological evidence of lipoblasts and lipocytes; the diagnosis is confirmed with immunohistopathological staining positive for murine double minute 2 (MDM2) or cyclin-dependent kinase 4 (CDK4) [9,10,11].

Magnetic-resonance imaging (MRI) is one of the most useful diagnostic tools for lipomatous soft-tissue tumors. Intramuscular (IM) lipomas have a homogeneous, high-signal intensity on T1-weighted (T1W) and T2-weighted (T2W) images. T1-weighted images are produced by using short ‘time to echo’ (TE) and ‘repetition’ (TR) time. Conversely, T2W images are produced by using longer TE and TR times. The contrast and brightness of other soft-tissue tumors are mostly of low signal intensity (dark) on T1W and high signal intensity (bright) on T2W; however, in lipomatous tumors, both T1W and T2W images are high in signal intensity (bright), as is the fat density. Their characteristic feature is that they show a low signal intensity (dark) on T1W/T2W images with fat suppression. The MRI features of these IM lipomas may resemble ALTs/WDLSs because of their larger size, deep location, and occasionally enhanced thick septa, whereas distinct high-grade liposarcomas can be easily identified by their heterogeneous high signal intensity on T2W images [12,13]. However, the inter-observer reliability of diagnosis in patients with lipomatous soft-tissue tumors showed only slight-to-substantial agreement [14,15]. If tumors present a diagnostic dilemma to the medical care team owing to the difficulty in distinguishing low-grade tumors (ALTs/WDLSs) from benign tumors (IM lipomas) from imaging alone, biopsy is considered as the reference standard. Although image-guided biopsy is a useful adjunct in evaluating the histological grade of lipomatous soft-tissue tumors, the procedure is invasive and biopsy sampling error can arise [16]. To help ease the diagnostic uncertainly following MRI, constructing a diagnosis system prior to surgery is very useful for treatment planning, especially in determining the urgency of surgery and avoiding insufficient or excessive therapy.

Artificial intelligence is a very useful diagnostic tool and it is becoming popular in evaluating the histological grade of cancer. Recent studies proposed MRI-based radiomic machine learning as a predictive model in distinguishing between lipomas and liposarcomas [15,17,18,19,20,21,22,23]. Tang et al. [22] and Yang, et al. [21] reported that performing machine learning in distinguishing between lipomas and ALTs/WDLSs on preoperative MRI showed greater precision than MSK radiologists using both T1W and T2W FS images in two dimensions (2D). Most of these studies included 2D, single T1W images or plus T2W FS images in order to achieve diagnostic accuracy. Recent advances in imaging retrieval allow segmentation in multiple 2D images to build up into three-dimensional (3D) images. However, none of the previous studies [15,17,18,19,20,21,22,23] used 3D images. The trade-off for a longer analysis time and CPU space made the 3D approach more labor-intensive and less desirable, even though 3D images may yield more details and provide better accuracy than 2D images, [24]. However, no previous studies confirmed the superior efficacy of 3D segmentation over 2D in diagnosing lipomatous neoplasms. The performance of the predictive model depends on the stability of the radiomic features. These features of a segmented region of interest can be extracted from 2D or 3D MRI images, which may yield stable or unstable features. The intra- and inter-observer variability (the variability in repeated segmentations by the same observer segmenting the same images on separate occasions and the variability in the segmentations between observers) in the manual segmentation of tumor regions may lead to unstable or low-robustness features. Therefore, radiomic -robustness evaluation prior to the selection process is important to ensure the precision of features. 

The primary research question in this study was: Are the radiomic features for 3D tumor segmentation more likely to be reproducible than those in the 2D approach? The study compared the effects of intra- and inter-observer manual-segmentation variability on feature reproducibility between 2D and 3D MRI-based radiomic features in lipomatous soft-tissue tumors. We also compared the efficacy of 2D and 3D contour-focused segmentation in terms of the robustness of the features, stable feature rate, sensitivity, specificity, and diagnostic accuracy of machine learning in the reproducible features.

## 2. Materials and Methods

### 2.1. Patients

This retrospective study was conducted after approval from the ethics committee of our institute, with a waiver of informed consent. The study was approved by our institutional review board (SIRB) to retrieve the MRI images and reveal the pathological report from the set of lipomatous-tumor patients. Furthermore, the board acknowledged that the new paraffin-embedded tissue blocks were freshly sliced from patients with both lipomas and well-differentiated liposarcomas for MDM2 and CDK4 immunohistochemistry were freshly sliced, in addition to the conventional treatment. Data set was obtained from 60 patients with lipomatous soft-tissue tumors (40 IM lipomas and 20 ALTs/WDLSs). Between 2010 and 2022, all cases underwent preoperative MRI scans and a complete tumor excision; subsequently, their diagnoses were confirmed by histopathology and immunohistochemistry or fluorescence in situ hybridization (FISH) examination. The tumors with and without murine double minute 2 (MDM2), and cyclin-dependent kinase 4 (CDK4) gene amplification showed by immunohistochemistry/FISH were diagnosed as ALT/WDLS and IM lipoma, respectively [11,25]. In the classification process, the final pathological results were considered as the gold standard in distinguishing between IM lipomas and ALTs/WDLSs.

### 2.2. MRI Data Sets

The preoperative MRI of the patients was acquired through a variety of scanners and sequences. In this study, we focused on the T1 weighted sequence (T1W) because it was the most readily available at our institution. The data sets included 46 axial (78.0%), 11 coronal (18.6%), and 2 sagittal (3.4%) planes of T1W MRI, all of which expose lipomatous soft-tissue tumors clearly. All images were acquired by 3.0 Tesla MRI scanners (Philips Healthcare, Netherlands). Median value of the MRI slice thickness was 5 mm (range, 3–8). Median values of the repetition time (TR) and echo time (TE) for T1WI sequence were 579 ms (range, 425–1425) and 15 ms (range, 7–25), respectively.

### 2.3. Two-Dimensional and Three-Dimensional MRI Image Segmentations

The open-source software, 3D Slicer, version 4.11.20210226 r29738/7a593c8 (https://download.slicer.org/, accessed on 21 April 2021) was used for manual segmentation of lipomatous soft-tissue tumors (Figure 1 and Figure 2). This software provides a set of interactive, easy-to-use tools that can be efficiently used for image analysis and scientific visualization. One module of the software is an interactive editor tool, which contains a variety of interactive segmentation effects [26,27,28]. 

All segmentations were performed by using a competitive region growing algorithm implemented in 3D Slicer, with a manual correction. Regarding N-class segmentation, the algorithm needs an N initial set of labeled pixels that corresponds to each class from the user. Next, the algorithm generates the region of interest (ROI), which is the convex hull of the user-labeled pixels with an additional margin. Subsequently, it iteratively labels all remaining pixels in ROI by using user-given pixel labels. The pixel labeling is performed using a weighted similarity score, which is a function of neighboring pixel weights. In addition, an unlabeled pixel is labeled corresponding to the neighboring pixels that have the highest weights. When all pixels in ROI have unchanged labels across several iterations, the algorithm converges [26,27,29] (Figure 3). 

In all cases, tumors were segmented on the original T1W MRI sequences: 

(1) Tumor ROIs were drawn on the slices showing the largest tumor area for 2D-image segmentation and the total tumor volume for 3D-image segmentation. 

(2) A reference ROI was drawn in fat on T1W image for the image-intensity-normalization procedure. 

All tumors were selected blindly and randomly for repeated segmented ROI by two observers (statistician and research scientist), after which their precision was confirmed by two experts (in musculoskeletal radiology and orthopedic oncology). For assessment of intra-observer variability, one observer repeated the segmentation of all cases after a pause of 2 weeks. As well as a second observer, all tumors were analyzed to evaluate inter-observer variability.

The next step was the image-intensity normalization. The different scanners or parameters that were used for the scanning of different patients, may have cause significant intensity variations. These variation would have significantly undermined the performance of subsequent MR image processing [30]. Consequently, the segmented *ROI* of each case was normalized to adjust for differences in *T*1*W* MRI protocols before radiomic feature extraction.

The normalization method is given below. 

The normalized intensity value (*NIV*) was:(1)$$NIV=\frac{Original\;T1W\;intensity}{Mean\;intensity\;value\;within\;the\;reference\;ROI}\times 1000$$

As a statistical measure of the dispersion of data points in a data series around the mean, the coefficient of variation (CV) is useful for comparing the degree of variation between one data series and another, even though, in our case, the means are significantly different from one another. This led us to the concept of Equation (1). The *NIV* is the ratio of the original *T*1*W* intensity to the mean intensity value within the reference *ROI*. 

### 2.4. Radiomic Feature Extraction

The radiomic features of tumor segmented ROI were extracted from the normalized T1W intensity using PyRadiomics version 3.0.1. PyRadiomics is an open-source Python package for extraction of radiomics data from medical images, with a simple and convenient front-end interface in 3D Slicer [31]. These features were classified as follows:

(1) Histogram-based features (i.e., first-order features) included 18 features, which were the simplest statistical descriptors based on single-pixel or single-voxel analyses that described the distribution of their gray-level values. 

(2) Texture-based features were subdivided into 5 classes: gray-level co-occurrence matrix (GLCM, 24 features); gray-level run-length matrix (GLRLM, 16 features); gray-level size-zone matrix (GLSZM, 16 features); gray-level dependence matrix (GLDM, 14 features); and neighboring gray-tone-difference matrix (NGTDM, 5 features). These features describe the correlation of neighboring pixel or voxel values and the homogeneity or heterogeneity of ROI [32,33]. 

(3) Shape-based 3D (14 features) features describe the geometric properties of ROI. 

The definition and formula of each feature were described according to PyRadiomics’ documentation. (https://pyradiomics.readthedocs.io/en/latest/features.html, accessed on 21 April 2021) [28,32,33,34,35,36] (Table A1).

### 2.5. Diagnotic Efficacy of MRI-Based Radiomic Features 

To appraise the diagnostic efficacy of MRI-based radiomic features that extracted from 2D and 3D tumor segmentation, a classification model was built based on supervised machine-learning. As a first step, we randomized the acquisition data into two subsets (80% of data were used for the learning set and 20% for the test set). Next, we performed a ten-fold cross-validation (9-fold for training set and 1-fold for validation set) on a learning set, which comprised two procedures (i.e., feature selection and constructed model). Both procedures were performed using least absolute shrinkage and selection operator (LASSO) logistic regression. This method is computationally feasible for high-dimensional data [37]. Eventually, the best model was selected and, subsequently, the sensitivity, specificity, and diagnostic accuracy of the predictive model on the test set were assessed [37,38] (Figure 4). 

For the machine-learning algorithm of model 3, first, we used the principal component analysis (PCA) to create the new variable, which combined 2D and 3D radiomic features. In the second step, support vector machine (SVM), which suitable for high-dimensional and small-sample-size data [39], was used to classify the soft-tissue-tumor differentiations, and the classifiers were validated using Monte-Carlo cross-validation (MCCV). This method is suitable for studies with limited sample sizes [40]. Finally, the sensitivity, specificity, and diagnostic accuracy of the predictive model on the test set were appraised [38].

### 2.6. Statistical Analysis

The descriptive statistics were used to describe the characteristics of the data. We compared the characteristics between IM lipomas and ALTs/WDLSs groups using the Student’s t-test for continuous data and the Chi-squared test for categorical data.

Intraclass correlation coefficient (ICC) was used to evaluate the effects of intra- and inter-observer manual-segmentation variability on feature reproducibility. In the calculation of ICC value, we chose a two-way random-effects model with absolute agreement [41]. The ICC interpretations were as follows: excellent (ICC ≥ 0.90); good (0.75 ≤ ICC < 0.90); moderate (0.50 ≤ ICC < 0.75); and poor (ICC < 0.50) [42]. In this study, if the ICC value was more than 0.75, a feature was considered reproducible [42,43,44,45,46,47]. This meant that the features were stable. Moreover, we compared the stable feature rate between two groups (2D vs. 3D tumor segmentation) using the Chi-squared test. 

We classified the degree of feature robustness into 3 categories: low (ICC < 0.50), moderate (ICC, 0.50–0.75), and high (ICC > 0.75) [42,47,48].

Statistical analysis and graphs were performed using R version 4.2.1 (R Foundation for Statistical Computing, Vienna, Austria). A *p* value that was less than 0.05 was considered statistically significant (Figure 2).

## 3. Results

Sixty patients were diagnosed, including 40 with IM lipomas and 20 with ALTs/WDLSs. Most were women (65.0%). Although the proportion of women in the IM-lipomas group was higher than that in the ALTs/WDLSs group, there was no statistically significant difference between the two groups (70% vs. 55%, *p* = 0.3). The average age of the patients was 60.2 years (SD, 9.5; range, 38 to 80). There was no statistical difference in the average ages of both groups (IM-lipomas group, 60.6 years vs. ALTs/WDLSs group, 59.7 years; *p* = 0.7). Most of the deep-seated soft-tissue tumors were in the thigh and arm/shoulder (Table 1).

In Table 2 and Table 3 and Figure 5, the ICC values were used to evaluate the intra- and inter-observer manual-segmentation variability. The criterion of ICC > 0.75 was considered as a stable feature.

For the intra-observer variability of the radiomic-features family, the ICC values for the 2D tumor segmentation ranged from 0.65 to 0.86, and for 3D tumor segmentation, they ranged from 0.77 to 0.97. The stable feature rate for the 2D and 3D tumor segmentations were 63.4% (59 of 93 features) and 78.5% (84 of 107 features), respectively. In the 2D vs. 3D tumor segmentation, there was a statistically significant difference in the stable feature rates of the two groups (*p* = 0.018) (Table 2) (Figure 5a).

Regarding the inter-observer variability of the radiomic-features family, the ICC values for the 2D tumor segmentation ranged from 0.42 to 0.70, and for the 3D tumor segmentation, they ranged from 0.79 to 0.92. For the 2D vs. 3D tumor segmentation, the stable feature rate of both groups was 46.2% (43 of 93 features) and 71.0% (76 of 107 features), which suggested a statistically significant difference between two groups (*p* < 0.001) (Table 3) (Figure 5b).

In Table 4 and Figure 6, we measure the robustness of the radiomic-features family using the ICC of both the intra- and the inter-observer variability. The degree of feature robustness included 37.6% high, 21.5% moderate, and 40.9% low for the 2D tumor segmentation and 58.9% high, 40.2% moderate, and 0.9% low for 3D tumor segmentation. Only 35 features were found (37.6%, 35 of 93 features) with ICC > 0.75 of both intra- and inter-observer variability for the 2D tumor segmentation and 63 features (58.9%, 63 of 107 features) were found for the 3D tumor segmentation. The stable feature rate for the 2D tumor segmentation was lower than that for the 3D tumor segmentation (37.6% vs. 58.9%, *p* = 0.002). 

The performance of the machine-learning models in terms of the reproducible features is shown in Table 5. The models achieved a diagnostic accuracy of 88% (model 1: sensitivity, 75% and specificity, 100%) for 2D tumor segmentation and a 90% diagnostic accuracy (model 2: sensitivity, 83% and specificity, 100%) for 3D tumor segmentation. For the 2D + 3D radiomic features, the model achieved a diagnostic accuracy of 87% (model 3: sensitivity, 100% and specificity, 80%).

## 4. Discussion

The clinical implementation of radiomics machine learning still faces challenges. Regarding lipomatous soft-tissue tumors, radiomic studies have so far focused on differentiating between benign and malignant tumors [15,17,18,19,20,21,22,23,49]. Despite the increasing development of the radiomic-machine-learning model, achieving robustness in the radiomic features extracted from MRI images is still a challenge in radiomic studies. Furthermore, the segmentation of tumors is also an important procedure. However, despite the robustness of MRI-based radiomic features, no study has yet been performed to compare the efficacy of 2D contour-focused segmentation with the 3D approach in lipomatous soft-tissue tumors or other soft-tissue tumors, as reported in cartilaginous bone tumors [50]. In our study, we analyzed the intra- and inter-observer variability to assess the performance of the segmentation method. Based on the results, the 2D contour-focused segmentation of lipomatous soft-tissue tumors has lower ICC values compared to the 3D approach. Although 2D and 3D MRI-based texture analyses provide similar robustness/stability of features, some features have low robustness because of the variability of the segmentation technique. Four sets from two observers are shown in Figure 5. It can be seen that both the 2D and the 3D tumor segmentations revealed significant variability in their radiomic features because the tumor region was drawn using the segmentation technique. The experience of the observers also affects the segmentation of tumors, which is shown by differences between radiomic features. However, 3D contour-focused segmentation shows high ICC values for both intra-and inter-observer effects compared to the 2D approach. This suggests that the radiomic features extracted from 3D tumor segmentation are more reproducible and stable/robust. In Table 6, we summarize the segmentation techniques and the robustness of MRI-based radiomic features, along with the stable feature rate, in lipomatous soft-tissue tumors and compare the strengths and weaknesses of previous and current studies. 

Although the performance of 3D contour-focused segmentation is superior to the 2D approach, it takes more time than the 2D approach to segment tumors. However, the 2D MRI-based radiomic features of lipomatous soft-tissue tumors are reproducible. In future studies, 2D contour-focused segmentation may be favored because it is easier to implement in the clinical practice. Therefore, to minimize the risks associated with using noisy, unstable, and unreproducible features in radiomic-image analysis, it is advisable to apprise the feature robustness of different segmentation approaches prior to conducting radiomic studies. 

In our study, there are two limitations. First, we investigated only the T1W sequence, since it was the most readily available at our institution. Second, the sample of 40 IM lipomas and 20 ALTs/WDLSs were relatively small, especially for diagnostic efficiency. However, our study was focused on the reproducibility and robustness of the radiomic features extracted from 2D and 3D MRI image segmentations. Thus, adapting machine learning, a further study should be conducted on a large scale with a larger sample size so that the application can be highly accurate. We will continually evaluate MRI scans of IM lipomas and ALTs/WDLSs to progressively refine our model. Small tumors treated with observation or masses that were not referred to the sarcoma center were excluded from this study.

## 5. Conclusions

The main findings of this study is that the stable rate/robustness of radiomic features extracted from the T1W MRI were 37.6%, or high robustness, for the 2D contour-focused segmentations and 58.9%, or high robustness, for the 3D contour-focused segmentations. The radiomic features of lipomatous soft-tissue tumors were extracted from 2D and 3D manual segmentations on the MRI images were reproducible. However, the 2D MRI-based texture analysis provided features that were less robust than those obtained using the 3D approach. Segmentation using the competitive region growing algorithm produces good reproducible features. Therefore, this algorithm should be further applied to the machine-learning model with more sample data, taken from multiple centers.

## Figures and Tables

**Figure 1 diagnostics-13-00258-f001:**
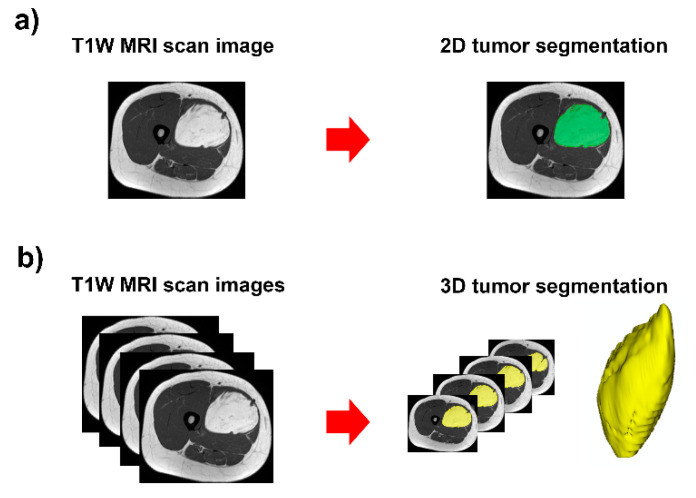
Contour-focused segmentation provides the region of interest of the tumor: (**a**) 2D tumor segmentation in green area; and (**b**) 3D tumor segmentation in yellow area.

**Figure 2 diagnostics-13-00258-f002:**
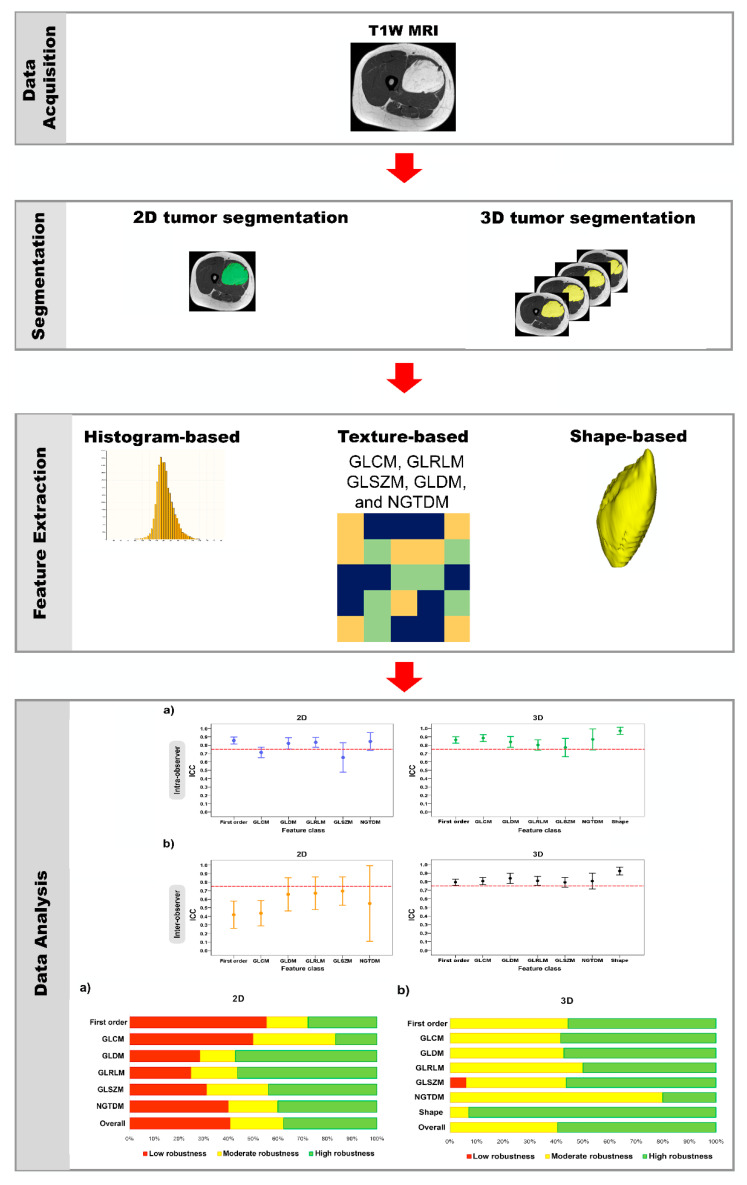
Schematic diagram of the method used in this study.

**Figure 3 diagnostics-13-00258-f003:**
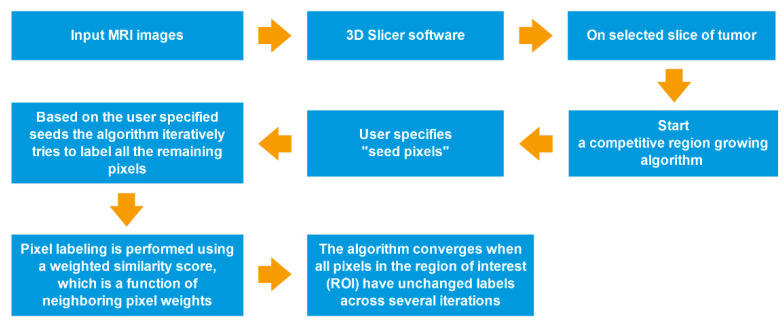
Process-flow diagram of a competitive region growing algorithm.

**Figure 4 diagnostics-13-00258-f004:**
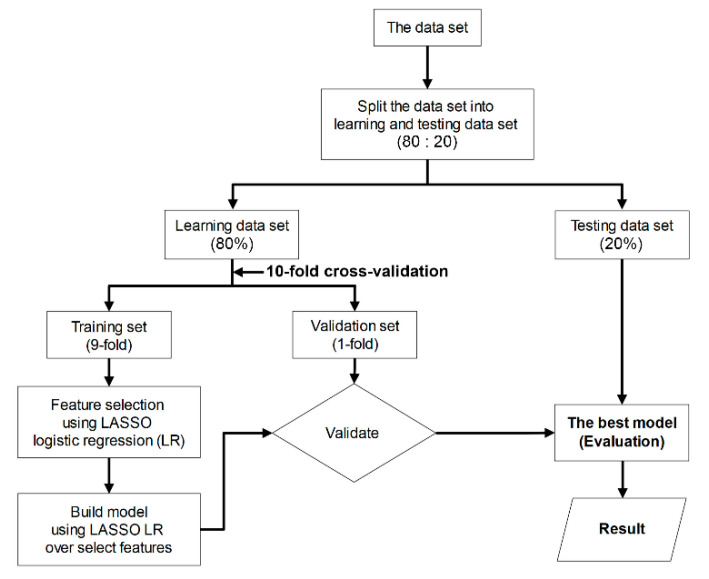
Flowchart of the machine-learning procedure used to evaluate the performance of classification models.

**Figure 5 diagnostics-13-00258-f005:**
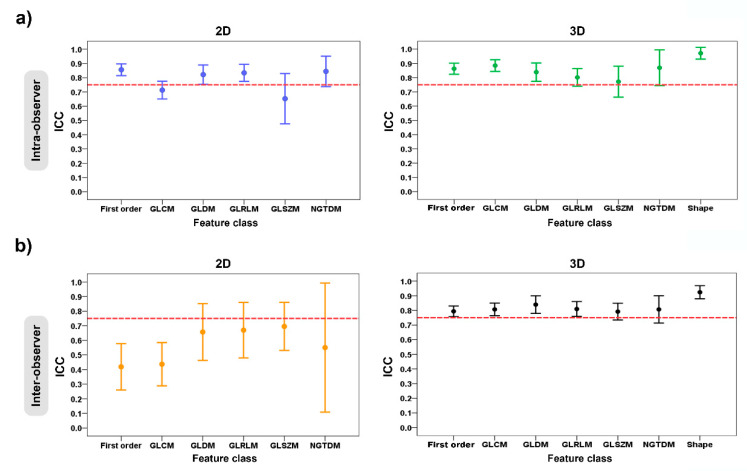
Error plots show the intra- and inter-observer variability of feature classes grouped according to 2D and 3D tumor segmentations: (**a**) intra-observer variability; and (**b**) inter-observer variability. The red dashed horizontal lines denote the intraclass correlation coefficient (ICC) value of 0.75.

**Figure 6 diagnostics-13-00258-f006:**
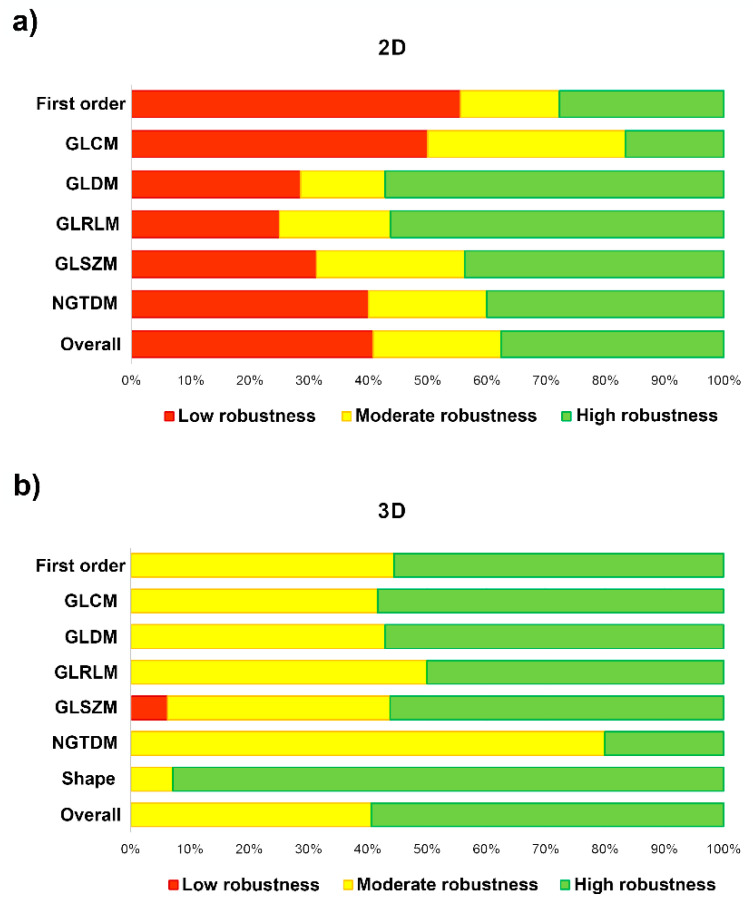
Bar-chart plots show the degree of feature robustness: (**a**) 2D tumor segmentation; and (**b**) 3D tumor segmentation.

**Table 1 diagnostics-13-00258-t001:** Characteristics of patients with lipomatous soft-tissue tumors.

Variable	Total Patients (n = 60)	IM Lipoma (n = 40)	ALTs/WDLSs (n = 20)	*p* Value
Gender				
male	21 (35.0%)	12 (30.0%)	9 (45.0%)	0.3
female	39 (65.0%)	28 (70.0%)	11 (55.0%)	
Age (year)	60.2 (9.5)	60.6 (9.1)	59.7 (10.3)	0.7
Location of tumor				
Thigh	29	15	14	
Arm	14	12	2	
Shoulder	10	10	0	
Leg	5	1	4	
Back/trunk	2	2	0	

IM lipomas, intramuscular lipomas; ALTs/WDLSs, atypical lipomatous tumors/well-differentiated liposarcomas.

**Table 2 diagnostics-13-00258-t002:** Intra-observer variability of the radiomic-features family and stable feature rate for 2D and 3D tumor segmentations.

Feature Class	2D Tumor Segmentation			3D Tumor Segmentation		
	Number of Total Features	ICC Value	Stable Feature Rate (%)	Number of Total Features	ICC Value	Stable Feature Rate (%)
First order	18	0.86 ± 0.09	77.8	18	0.86 ± 0.08	83.3
GLCM	24	0.71 ± 0.15	37.5	24	0.88 ± 0.10	83.3
GLRLM	16	0.83 ± 0.12	81.3	16	0.80 ± 0.12	68.8
GLSZM	16	0.65 ± 0.35	56.3	16	0.77 ± 0.22	68.8
GLDM	14	0.82 ± 0.13	71.4	14	0.84 ± 0.12	78.6
NGTDM	5	0.84 ± 0.12	80.0	5	0.87 ± 0.14	60.0
Shape	-	-	-	14	0.97 ± 0.08	92.9
Overall	93	0.77 ± 0.20	63.4	107	0.86 ± 0.14	78.5

ICC, intraclass correlation coefficient; the resulted ICC values are represented using mean ± standard deviation.

**Table 3 diagnostics-13-00258-t003:** Inter-observer variability of the radiomic-features family and stable feature rate for 2D and 3D tumor segmentations.

Feature Class	2D Tumor Segmentation			3D Tumor Segmentation		
	Number of Total Features	ICC Value	Stable Feature Rate (%)	Number of Total Features	ICC Value	Stable Feature Rate (%)
First order	18	0.42 ± 0.34	27.8	18	0.79 ± 0.09	72.2
GLCM	24	0.44 ± 0.36	25.0	24	0.81 ± 0.11	70.8
GLRLM	16	0.67 ± 0.38	62.5	16	0.81 ± 0.10	68.8
GLSZM	16	0.70 ± 0.33	62.5	16	0.79 ± 0.11	62.5
GLDM	14	0.66 ± 0.36	64.3	14	0.84 ± 0.11	64.3
NGTDM	5	0.55 ± 0.49	60.0	5	0.81 ± 0.10	60.0
Shape	-	-	-	14	0.92 ± 0.08	92.9
Overall	93	0.56 ± 0.37	46.2	107	0.82 ± 0.11	71.0

ICC, intraclass correlation coefficient; the resulting ICC values are represented using mean ± standard deviation.

**Table 4 diagnostics-13-00258-t004:** Robustness of the radiomic-features family and stable feature rate for 2D and 3D tumor segmentations.

Feature Class	2D Tumor Segmentation				3D Tumor Segmentation			
	Total Features	Low Robustness	Moderate Robustness	High Robustness	Total Features	Low Robustness	Moderate Robustness	High Robustness
First order	18	10	3	5	18	0	8	10
GLCM	24	12	8	4	24	0	10	14
GLRLM	16	4	3	9	16	0	8	8
GLSZM	16	5	4	7	16	1	6	9
GLDM	14	4	2	8	14	0	6	8
NGTDM	5	2	1	2	5	0	4	1
Shape	-	-	-	-	14	0	1	13
Overall	93	38	20	35	107	1	43	63
Stable feature rate	-	-	-	37.6% (35/93)	-	-	-	58.9% (63/107)

ICC, intraclass correlation coefficient; both intra- and inter-observer variability (with ICC) were used to assess feature robustness.

**Table 5 diagnostics-13-00258-t005:** Comparing diagnostic accuracy of machine-learning models between 2D and 3D tumor segmentations.

Machine Learning	Features	Number of Reproducible Features	Number of Selected Features in Final Model	Sensitivity (%)	Specificity (%)	Accuracy (%)
Model 1	2D radiomic features only	35	13	75	100	88
Model 2	3D radiomic features only	63	10	83	100	90
Model 3	2D + 3D radiomic features	98	4	100	80	87

**Table 6 diagnostics-13-00258-t006:** Summary of the literature and current study for MRI-based radiomic-feature robustness in lipomatous soft-tissue tumors.

Study	Segmentation Technique	Stable Feature Rate	Strengths	Weaknesses
Cay, 2022 [23]	- 2D approach - Manual segmentation with the free-hand method	87.8%	- Measures the robustness of radiomic features using inter-observer reproducibility	- The study only consists of T1-weighted turbo-spin echo (T1W TSE) sequence - Sample data (n = 65) - There is no assessment of intra-observer manual-segmentation variability
Fradet, 2022 [49]	- 3D approach - Manual segmentation	N/A	- The sample size (n = 145) in this study is larger than that in any previous studies	- The study only comprises fat-suppressed gadolinium contrast-enhanced T1W sequence - Intensity-distribution features were extracted from masked MRI images without normalization - There is no radiomic feature-robustness evaluation
Yang, 2022 [21]	- 3D approach - Manual segmentation	N/A	- The study consists of T1WI and T2FS sequences - Measures the robustness of radiomic features using intra-observer variability - Sample data (n = 127)	- Tumor-segmented ROI for each case was not normalized before radiomic-feature extraction - There is no assessment of inter-observer manual segmentation variability - Only eighteen patients (20% of 89) were used for the intra-observer manual-segmentation variability assessment
Tang, 2022 [22]	- 3D approach	T1W, 42.3%; FS T2W 25.0%; and T1W & T2W, 39.4%	- The study consists of T1WI and fat-suppressed T2-weighted sequences - Tumor segmented ROI for each case was normalized before radiomic-feature extraction - Measures the robustness of radiomic features using inter-observer variability - Sample data (n = 122)	- There is no assessment of intra-observer manual-segmentation variability
Pressney, 2020 [18]	- 2D approach - Manual segmentation	N/A	- The study consists of five MRI sequences.	- Small sample size (n = 30) - Segmentation was performed only in two dimensions
Leporq, 2020 [17]	- 2D approach - Manual segmentation	63.2%	- Measures the robustness of radiomic features using inter-observer variability with Pearson’s correlation coefficient (r)	- The study only consists of gadolinium-contrast enhanced T1W sequence - Tumor segmented ROI for each case was not normalized before radiomic-feature extraction - There is no assessment of intra-observer manual segmentation variability - Segmentation was performed only in 2D - Sample data (n = 87)
Malinauskaite, 2020 [15]	- 3D approach - Automatic segmentation	76.1%	- Measures the robustness of radiomic features using inter-observer variability	- The study only consists of T1W sequence. - Tumor segmented ROI for each case was not normalized before radiomic-feature extraction - There is no assessment of intra-observer variability - Only twelve patients were included for the inter-observer variability evaluation
Vos, 2019 [20]	- 3D approach - Semi-autosegmentation	N/A	- Tumor segmented ROI for each case was normalized before radiomic-feature extraction - Sample data (n = 122)	- There is no radiomic-feature-robustness evaluation
Thornhill, 2014 [19]	- 3D approach	N/A	- The study consists of three MRI sequences - Tumor segmented ROI for each case was normalized before radiomic-feature extraction	- There is no radiomic-feature-robustness evaluation - Small sample size (n = 44)
Current study	- 2D approach	37.6%	- Tumor segmented ROI for each case was normalized before radiomic-feature extraction - Measures the robustness of radiomic features using ICC of both intra- and inter-observer variability	- The study only comprises T1W sequence. - The study uses a limited sample size (n = 60)
	- 3D approach - Segmentation with a competitive region growing algorithm	58.9%		

## Data Availability

Data available upon request from the authors.

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
