# Peer review of "Robustness of Radiomic Features: Two-Dimensional versus Three-Dimensional MRI-Based Feature Reproducibility in Lipomatous Soft-Tissue Tumors"

_diagnostics, 2023, doi:10.3390/diagnostics13020258_

Round 1
Reviewer 1 Report
The work is of scientific interest to the research community, as it addresses a topic that greatly affects the world's population, such as cancer. The article is well structured, with some minor changes that can be made and describes in a general way the procedure used. However, it is not possible to identify the scientific contribution of the research. Although the authors state that they will make a comparison, they do not say if the research is new in that sense, or if there are previous works, which is what they contribute with this one. A description of what they are introducing, rather than selecting some machine learning models and extracting the results, would be desirable.
Next issues should also be addressed:
1. Page 1, lines between 40 and 45, more description about lipomatous tumors should be provided, what characterizing it, risk, prognosis, epidemiology?
2. What differentiates T1W from T2W? Description from both should be provided. Line 41.
3. An in deep review of the SOTA should be placed in the introduction, you write “Several studies in recent years pro-57 posed the MRI-based radiomic machine learning as predictive model…” but only 5 references are given. Lines between 57 and 59.
4. A brief summary of the SOTA analysis should be added in the Introduction section, showing successes in previous research and areas for improvement where your work can add value.
5. The contribution of this study should be clearly stated in the introduction, before the final paragraph.
6. What is the difference between intra- and interobserver variability? Describe each one. Line 62-63.
7. Under what rules is your institution's ethics committee governed? Provide information. Page 2, line 74.
8. Patient consent was obtained to analyze their images. Page 2, line 75.
9. Abreviation should be defined in line 80 (MDM2, CDK4, P-16).
10. Figures must be placed right after mention in line 95, page 3.
11. A diagram of the process described between lines 97 and 104 on page 3 should be provided in other to keep readers attention in a graphical way.
12. PyRadiomics and 3D Slice software, are proprietary?, if so, have their license of use, if they are open source provide creators, characteristics that made them be selected. Lines 94 and 125, page 3.
13. The mathematical formula utilize can be provided in this paper for a robust description of features. Line 136.
14. No prior match or description at the time of calling Figure 2. Lines 137-138.
15. In figure 2, graphs information can’t be appreciated in Data Analysis section, line 143. Improve image quality.
16. Models and procedures utilized must be explained in details, showing why are there suitable for the task descripted (LASSO Logistic Regresion, SVM, Monte-Carlo cross validation).
17. Statistical analysis section should be placed after Diagnosis efficacy of MRI-based radiomic features section, in order to keep a flow based on feature extraction of section 2.4, models applied and statistical analysis then.
18. The discussion section should be more precise about why the results were obtained, and not just repeat what the results showed, which is evident when reading them. For example, what is the reason for the variability in the segmentation techniques, which introduced low robustness in some characteristics, which characteristics make them different from each other. What can be done to improve them?
19. The limitations of your study, if any, should be added in the conclusions.
20. Please highlight your contributions in the conclusion section.
21. It looks a comprehensive literature survey is missing in the article, authors are suggested to add more latest references.
Author Response
Dear Reviewer 1,
On behalf of my co-authors, we have revised our manuscript “Robustness of radiomic features: 2D versus 3D MRI-based feature reproducibility in lipomatous soft tissue tumors” (Manuscript ID: diagnostics-2086670) according to the editors and reviewers’ recommendation.
We would like to thank you for the thoughtful comments and helpful suggestions. We fundamentally agree with all the comments, and we have incorporated corresponding revisions into the manuscript. We now submit my revised manuscript as attachment file for consideration as a research article for possible publication in Diagnostics.
For our detailed, point-by-point responses to the editorial and reviewer comments are given as attachment file, whereas the corresponding revisions are marked up using the “Track Changes” function in MS Word, in the manuscript file.
We believe that our manuscript has been considerably improved as a result of these revisions, and hope that our revised manuscript “Robustness of radiomic features: 2D versus 3D MRI-based feature reproducibility in lipomatous soft tissue tumors” is acceptable for publication in the Diagnostics.
We would like to thank you once again for your consideration of our work.
Yours sincerely
Authors

Reviewer 2 Report
This a good technical paper however some parts need improvement especially the literature review and results & discussion.
1. The introduction section needs to be improved with more previous related researches. Related work/research line involved in related applications are not comprehensively discussed.
What are the research gap / issues / weaknesses of those researches and how your researches can solve those issues.
2. Then, please discuss the contribution of this paper.
3. Please add the table of research comparison of the previous researches by stating the strengths and weaknesses of each research.
4. The equation in line (119) has no numbering. How do you formulate this equation and please explain briefly.
5. The results are excellent. How ever, please add result comparison from the most related researches which have been discussed in the introduction part. This is to support the effectiveness of your propose method.
6. Please revise the conclusion based on observation discussed in 5.
Author Response
Dear Reviewer 2,
On behalf of my co-authors, we have revised our manuscript “Robustness of radiomic features: 2D versus 3D MRI-based feature reproducibility in lipomatous soft tissue tumors” (Manuscript ID: diagnostics-2086670) according to the editors and reviewers’ recommendation.
We would like to thank you for the thoughtful comments and helpful suggestions. We fundamentally agree with all the comments, and we have incorporated corresponding revisions into the manuscript. We now submit my revised manuscript as attachment file for consideration as a research article for possible publication in Diagnostics.
For our detailed, point-by-point responses to the editorial and reviewer comments are given as attachment file, whereas the corresponding revisions are marked up using the “Track Changes” function in MS Word, in the manuscript file.
We believe that our manuscript has been considerably improved as a result of these revisions, and hope that our revised manuscript “Robustness of radiomic features: 2D versus 3D MRI-based feature reproducibility in lipomatous soft tissue tumors” is acceptable for publication in the Diagnostics.
We would like to thank you once again for your consideration of our work.
Yours sincerely
Authors

Reviewer 3 Report
The study is good, important, and has been addressed well. It needs some revisions.
1. Write the most important key contributions clearly.
2. The previous studies section should be added to discuss the techniques and results of at least 15 previous studies.
3. A figure methodology should be added for the study. It shows the tracking of the working mechanism from inputting images to output.
4. Write the equation on line 119 by typing, not as a picture.
5. Explain why normalization is used, with further explanation.
6. What are the limitations faced by the authors?
7. The results of the study should be compared with previous studies.
Author Response
Dear Reviewer 3,
On behalf of my co-authors, we have revised our manuscript “Robustness of radiomic features: 2D versus 3D MRI-based feature reproducibility in lipomatous soft tissue tumors” (Manuscript ID: diagnostics-2086670) according to the editors and reviewers’ recommendation.
We would like to thank you for the thoughtful comments and helpful suggestions. We fundamentally agree with all the comments, and we have incorporated corresponding revisions into the manuscript. We now submit my revised manuscript as attachment file for consideration as a research article for possible publication in Diagnostics.
For our detailed, point-by-point responses to the editorial and reviewer comments are given as attachment file, whereas the corresponding revisions are marked up using the “Track Changes” function in MS Word, in the manuscript file.
We believe that our manuscript has been considerably improved as a result of these revisions, and hope that our revised manuscript “Robustness of radiomic features: 2D versus 3D MRI-based feature reproducibility in lipomatous soft tissue tumors” is acceptable for publication in the Diagnostics.
We would like to thank you once again for your consideration of our work.
Yours sincerely
Authors

Round 2
Reviewer 3 Report
The authors answered my questions well.